# Women's Rights in Nigeria's Indigenous Systems: An Analysis of Non-Discrimination and Equality under International Human Rights Law

**Foluke Oluyemisi Abimbola [1], Stanley Osezua Ehiane [2,3],* and Roman Tandlich [4]**

[1] Faculty of Law, Lead City University, Ibadan 110115, Nigeria
[2] Department of Politics and Administrative Studies, Faculty of Social Sciences, University of Botswana, Gaborone 00704, Botswana
[3] School for Public Management and Governance, College of Business and Economics (CBE), University of Johannesburg, Auckland Park, P.O. Box 524, Johannesburg 2006, South Africa
[4] Division of Pharmaceutical Chemistry, Faculty of Pharmacy, Rhodes University, Grahamstown 6140, South Africa
* Correspondence: ehianes@ub.ac.bw or stanleyehiane@yahoo.com

**Abstract:** The Nigerian legal system is diverse in that it recognizes several established legal systems that regulate how Nigerians conduct themselves. These legal frameworks include the civil law that was passed down from the British during and after colonization, pre-colonial customary laws and cultural practices, and religious laws (Christian and Islamic laws). Different kinds of norms and laws have subjected Nigerian women to violations of their rights depending on the woman's cultural or religious affiliation. Such cultural and/or religious practices are usually in opposition to civil law and the Nigerian constitution, which is a custodian of these rights. Moreover, despite the supremacy of the constitution and expected compliance with international human rights treaties that Nigeria has ratified, the fact is that today there are impediments to the effective protection of women's rights in Nigeria. For instance, although the Nigerian constitution outlaw's discrimination on the grounds of gender, customary and religious laws continue to restrict the effective implementation of women's rights, making it extremely difficult to harmonize domestic legislation with international human rights conventions, and also remove discriminatory measures. This article, thus, examines the issues of gender inequality as the basis for agitation for women's empowerment and women's rights while also proposing a re-alignment of domestic legislation in compliance with international human rights conventions and treaties, in order to combat cultural and religious norms that flout human rights considerations for Nigerian women. Therefore, the main objective of this paper is to highlight the challenges that may arise when these legal systems clash, and how that affects the protection of women's rights, particularly in view of international human rights treaties which Nigeria has signed and ratified. The article will therefore propose that women's rights should be protected by seeking to eradicate cultural and religious practices that are discriminatory. This can be achieved by adopting laws which can be interpreted by domestic courts in line with constitutional requirements protecting the rights of women. It is noteworthy that the Nigerian judiciary has declared certain customs and traditions contrary to natural justice, equity and good conscience. Some of the case laws and judicial pronouncements will also be examined in this paper to enable implementation for the protection of women's rights. The methodology adopted is desk-top legal research where judgments of courts and legislative enactments will form the basis of the findings of this paper.

**Keywords:** gender; discrimination; customary law; legal pluralism in Nigeria; indigenous; law; rights; women

## 1. Introduction

The Federal Government of Nigeria is a signatory to the Convention on the Elimination of all forms of Discrimination against Women (CEDAW). Nigeria signed the convention on

23 April 1984 and, without any reservation, ratified the same on 13 June 1985. The optional protocol to CEDAW was also signed and ratified by Nigeria in 2004. As a result, Nigeria is committed to building an egalitarian society where every Nigerian enjoys equal rights irrespective of age and sex. In spite of these improvements, some states in Nigeria have only passed portions of CEDAW into law due to cultural and religious considerations. This situation has limited the implementation of CEDAW and other international human rights conventions in Nigeria with respect to women's rights. This plurality of laws has remained a challenge to the enforcement of women's rights as prescribed by the Nigerian constitution and other international conventions.

Nigeria's indigenous systems consist of cultural norms and societal beliefs that have gained widespread acceptance and recognition. This is referred to as customary law, where cultural perspectives are usually held on to tightly, so much so, that people in a community consider such norms mandatory. In Nigeria, cultural perspectives are well recognized among the people, such that over time, the Nigerian courts have taken judicial notice of such customs and traditions. Moreover, frequently followed by the courts. For instance, in the case of *Akinnubi v Akinnubi*, the Supreme Court of Nigeria held that 'it is a well-settled principle of native law and custom of the Yoruba that a wife could not inherit her husband's property.' This presumes that the Supreme Court has taken judicial notice of this Yoruba customary law. However, the existence of customary law has to be proved before the court can act on it. Thus, customary law is one of the recognized legal systems in Nigeria. It is defined as those rules of conduct which persons living in a particular locality have come to recognize as governing them in their relationships with one another (Kolajo 2000). In addition, customary law has been defined by Obaseki JSC as:

> "The organic and living law of the indigenous people of Nigeria regulating their lives and transactions. It is organic in that it is not static and is regulatory in that it controls the lives and transactions of the community subject to it. It is said that custom is the mirror of the culture of the people."

Usually proof that a particular custom is in existence in a court of law is important to the effective adjudication of the case where it is pleaded. The courts, however, exercise their discretion by considering statutory laws where it can be seen that the customary law creates untold hardship and/or is discriminatory. Thus, judicial intervention in such matters is inevitable in seeking to ensure equality and non-discrimination in the implementation of customary laws as the Nigerian legal system consists of statutes (enacted by the legislative arm of government), the common law (inherited from the United Kingdom as a result of colonialism), and customary law.

The issue of gender inequality, as the basis for agitation for women's empowerment and women's rights, has become critical in the sense that Nigeria is a male dominated society, where gender relations are based largely on patriarchal considerations (Sokefun 2010). Nevertheless, Articles 2(f) and 5 of the Convention on the Elimination of all forms of Discrimination against Women (CEDAW) require states 'to take all appropriate measures including legislation to modify or abolish existing laws, regulations, customs, and practices which constitute discrimination against women.' This is supported by Section 21 of the 1999 Nigerian Constitution which provides that:

> The state shall protect, preserve, and promote the Nigerian cultures which enhance human dignity and are consistent with the fundamental objectives as provided in this Chapter; and (b) encourage development of technological and scientific studies which enhance cultural values.

In addition, Section 42 (1) of the 1999 constitution denounces discrimination and states that:

> A citizen of Nigeria of a particular community, ethnic group, place of origin, sex, religion or political opinion shall not, by reason only that he is such a person (a) be subjected either expressly by, or in the practical application of, any law in force in Nigeria or any executive or administrative action of the government,

to disabilities or restrictions to which citizens of Nigeria of other communities, ethnic groups, places of origin, sex, religion or political opinions are not made subject.

In view of the above legal provisions, the objective of this study is to examine the relationship between culture and the realization of the human rights of women in Africa. The intention of this paper is not to address all cultural issues, but to consider some cultural obstacles and threats that have caused major setbacks, and examine ways of eradicating such obstacles from a legal and a socio-legal perspective. The prevalence of prejudicial traditional practices and customs that legitimize women's inequality in most African states hamper the effective implementation of human rights generally, and the rights of women as a vulnerable group in particular. Some of the obstacles/barriers to the realization of civil and political rights (CPR) and the economic, social and cultural rights (ESCR) of women are traditional practices and customs, which disproportionately affect realization of these rights, as traditionally this category of rights has often been marginalized rather than prioritized (Oloka-Onyango 2003). Barriers include lack of education for the girl-child where it is believed that a girl should be married off at puberty. It is common to find that young girls who marry as teenagers are unable to complete their education. They are usually disempowered. In the global struggle that identified the plight of uneducated girls in poor countries as most vulnerable, Nigeria holds the world's record of having the highest numbers of children out of school. The number is above ten million (Oduah 2018). Another cultural practice is female genital mutilation, which has led to the premature death of many women, and has also affected their reproductive rights (Okeke et al. 2012). In addition, discrimination in the workplace happens when women are paid less, because they have to embark on their maternity leave. This practice is contrary to national and international labor laws.

This paper concludes with observations and recommendations to mitigate these challenges, so that African states can comply with their domestic, regional, and international human rights obligations, in relation to the protection and promotion of women's human rights. Despite the above provisions, the Nigerian perspective on inheritance, marriage and family relations, participation in politics, and decision making is that the male gender is protected and aided and abetted by custom as the superior sex (Ekhator 2019). Social values and perceptions, which acknowledge practices that deny women and girls the opportunity of attaining relevance in various aspects of life, further lend weight to the general marginalization of women, shaping societal behavior towards them (WILDA 2002). Thus, it is imperative that customary beliefs and laws are subject to judicial review where such customs are brought before the courts and as a result abolished where necessary by the various organs of state, in compliance with international human right provisions. For instance, in the case of *Mojekwu v Mojekwu*, Niki Tobi, Justice of the Court of Appeal, declared that the *oli-ekpe* custom, which exists under the Nnewi Native Law in the eastern part of Nigeria, and which allows a male relative to inherit the property of a deceased person, if the deceased did not have a male child, is contrary to natural justice, equity, and good conscience. *Oli-ekpe* means "he inherited the property of his relation". In this case, the deceased died without writing a will but was survived by his widow and two daughters. The person that instituted the case in court (the plaintiff) was the son of the deceased's brother who sued for the title to the deceased's land in Onitsha (East of Nigeria) to the exclusion of the deceased's daughters and widow. The widow resisted the claim, contending that it is a violation of their human rights for her deceased husband's brother to inherit what should ordinarily be her children's. Contrary to this custom, the Court of Appeal (Enugu Division) upheld the right of the widow and the daughters to inherit their late husband and father's property. One of the Justices of the Court of Appeal, Niki Tobi J.C.A, emphasized in the judgment that:

> "Nigeria is an egalitarian society where the civilized sociology does not discriminate against women. However, there are customs all over which discriminate against the womenfolk which regard them as inferior to the menfolk. That should

not be so as all human beings, male and female are born into a free world and are expected to participate freely without any inhibition on grounds of sex. Thus, any form of societal discrimination on grounds of sex, apart from being unconstitutional is antithesis to a society built on the tenets of democracy. The *oli-ekpe* custom which permits the son of the brother of a deceased person to inherit his property to the exclusion of his female child, is discriminatory and therefore inconsistent with the doctrine of equity."

## 2. Nigerian Customary Laws and Practices in Comparison with Statutory Provisions and International Conventions

Perceptions and beliefs about women in Nigeria, which result in hardship and contravention of their rights as human beings, are usually tackled through judicial intervention and domestic legislation. The contradiction between customary laws/practices and statutory provisions in Nigeria arises when a law has been enacted or a provision is made in the Constitution, which addresses and denounces a cultural practice. The definition of customary law stated above expresses the fact that when a practice has attained some level of notoriety or popularity, it is usually difficult to change the mindset of the people seeking to enforce it. Hence, Nigerian children, especially the girl-child, are vulnerable to a wide range of abuse and harmful traditional practices. The national legal framework for child protection is the Child Rights Act 2003. To date, however, only 23 out of 36 states in Nigeria have adopted the Act. Implementation is patchy, with many local authority bodies unaware of their duties under the law. A national survey in 2014 found that 6 out of 10 children reported having suffered one or more forms of violence before reaching 18 years of age, with 70 per cent of them experiencing multiple incidents of violence. The country has the largest number of child brides in Africa: 23 million girls and women were married as children.

At 27 per cent, the prevalence of female genital mutilation/cutting (FGM/C) among girls and women aged 15–49 years is lower than in many countries where the practice is carried out. Nonetheless, Nigeria has the third highest absolute number of women and girls (19.9 million) who have undergone FGM/C worldwide. It is more commonly practiced in the south, pushed by grandmothers and mothers-in-law, aiming to curb promiscuity, prepare girls for marriage, and conform to tradition. These issues will be examined in three ways: firstly, by having judicial pronouncements on cultural beliefs, marriage, and inheritance; secondly, relating provisions of the 1999 Nigerian Constitution and legislative enactments to customary law and practices; and thirdly, making international conventions binding on Nigeria in addressing cultural/discriminatory practices.

### 2.1. Judicial Pronouncements on Cultural Beliefs, Marriage and Inheritance Constituting Inequality and Discrimination

2.1.1. Marriage

Marriages in Nigeria are contracted under three basic systems of marriage. These are: marriage under the Marriage Act (civil marriage), marriage under Customary Law, and Islamic law marriage. Marriage under the Marriage Act refers to a marriage contracted pursuant to the Marriage Act, which is an English Act first enacted in Nigeria as a British colony on the 31st of December 1914.

Marriage under customary law and Islamic law permit monogamous and polygamous marriages respectively. Conflicts arise when a woman is married to a man under the Marriage Act but their marriage affairs are carried out according to customary law. For instance, the Administration of Estates Law of the various states in Nigeria (one of the statutes now regulating inheritance rights in Nigerian societies), protects a wife who is married under the Marriage Act by ensuring that the wife is entitled to at least one-third of her husband's property upon death. In certain cases, the wife can be co-administrator of her late husband's estate. This provision is in conflict with most customs in Nigeria. Therefore, the status of the Nigerian woman and girl-child in the family is not sufficiently protected, and several inequalities, mainly due to tradition, remain. Another dimension is

the common knowledge that polygamy is prohibited in civil marriages but allowed under customary and Islamic laws. The practice is widespread, with more than one-third of Nigerian women in polygamous unions, even where the first marriage is contracted under the Marriage Act. This happens even though bigamy is an offence in the Nigerian criminal code. This disregard of statutory enactments, in furtherance of cultural beliefs, subjugates women under customary law, making them dependent on a system that treats them as less deserving of basic social and economic rights.

### 2.1.2. Child Marriage

Child-marriage is another harmful discriminatory custom against women and girls. In some regions of northern Nigeria, customary law permits girls to marry from the age of nine years. Records show that Nigeria has 40 percent of global child marriages, with 76 percent of adolescent girls married in the North-West, 68 percent in the North-East, and 35 percent, in the North-Central region, where incidentally, poverty is highest (Uwais 2017). However, the age of majority under the Nigerian Child Rights' Act is 18 years. The Act prohibits child marriage by providing that 'no person under the age of eighteen years is capable of contracting a valid marriage, and accordingly, a marriage so contracted is null and void and of no effect whatsoever.' The Act further states that 'no parent, guardian or any other person shall betroth a child to any person.' Even though the Act criminalizes child marriage in Section 23, child marriage is still a cultural practice that has continued unchecked (Fayokun 2015). Several communities, in a bid to justify the practice, claim that child marriage will ensure that child brides do not engage in sexual relations until puberty. On the contrary, early marriage itself often means a pre-mature entry into sexual activity. This can result in exposure to sexually transmitted diseases like HIV, as well as actual physical harm, and the high health risks of early pregnancy by a yet-to-mature girl. This clearly violates the rights to health and education. For instance, in a study carried out by Adedokun (Adedokun et al. 2016), data revealed that respondents commenced child bearing between 14 and 18 years of age, and 71% had experienced at least one serious pregnancy, or birth-related health problem, which include excessive bleeding during labor (19.0%), obstructed and/or prolonged labor (49.0%), frequent miscarriage (12.0%), and prolonged sickness after childbirth (20%). The same study revealed the comments of a nurse in Gombi General Hospital in Adamawa state, Nigeria, where young married girls presented for medical attention: "Most of the cases we witness in the health facilities are prolonged labor, Vesicovaginal fistula (VVF), and excessive bleeding. Some of these women are too young for child bearing but because of our culture and religion there is nothing we can do; we are educating them, but we still have new cases every day".

### 2.1.3. Inheritance

The disagreement in the three types of laws poses problems for inheritance matters. Women again are on the receiving end. Hence, the Nigerian superior courts have, through their decisions, set aside customs and traditions that have caused difficulties for women and widows. In terms of structure, the Nigerian legal system of court administration and hierarchy operates at three levels. At the apex, is the Supreme Court, which receives cases on appeal from the various Courts of Appeal divisions located all over the country. Whereas the Court of Appeal is also a federal court, it also has divisions located all over the country for administrative convenience. The lower-level courts are the High Courts of the various states in Nigeria (these are State courts, and they are at the apex of the State Courts, Nigeria being a Federation of States under the 1999 Constitution). These three tiers of courts are generally called Superior Courts of Record in contra-distinction to other lower-level courts not created by the Constitution, but by lesser statutes (for example, Magistrate Courts, Area Courts and Customary Courts).

Some cases have been decided by the Superior Courts declaring customs discriminatory. For instance, in the case of *Anekwe v Nweke*, the respondent, a widow, who had six daughters with her late husband, was ordered by the first appellant (her late husband's

nephew) to vacate the matrimonial home she shared with her late husband because she did not bear any male child in that marriage. The appellant's claim was based on the *Awka* custom that discriminates against daughters and widows from inheriting under intestacy. The appellant's claim was dismissed unanimously, and the Supreme Court declared that "the custom of *Awka* people of Anambra state to the effect that a married woman without a male issue cannot inherit the landed property of her late husband, pleaded and relied on by the appellants in the instant case, is barbaric and repugnant to natural justice, equity, and good conscience, and ought to be abolished".

In addition, the enactment of statutes can protect the inheritance rights of a widow. Therefore, if a deceased person was not legally married according to the terms of the Marriage Act, customary law may control how his inheritance is divided when he passes away intestate. However, if the marriage was performed in accordance with the Marriage Act, the deceased person's assets may be dispersed in accordance with the law, which only recognizes marriages performed according to the provisions of the Act. The Administration of Estates Law of 1959 governs the intestate succession of a person married under the Marriage Act during the deceased's lifetime in the South-Western states. Nonetheless, most states in Nigeria have enacted this law at different times and thus have more recent versions of it. It is significant to remember that the 1959 law's requirements do not apply in situations where customary law controls how an estate is distributed, inherited, and succeeded. This is so because, in accordance with the Administration of Estates Law of 1959, marriages that follow the Marriage Act are the only circumstances in which the distribution of an intestate estate is applicable. Real and personal property are distributed in detail under Section 49 in the event of intestacy. According to Section 49 (5):

> Where any person who is subject to customary law contracts a marriage in accordance with the provisions of the Marriage Ordinance and such person dies intestate after the commencement of this law, leaving a widow or husband or any issue of such marriage, any property of which the said intestate might have disposed by will shall be distributed in accordance with the provisions of this law, any customary law to the contrary notwithstanding.

Therefore, it has been established from the above provision of the law that under the statutory laws of succession, a woman married to a man under the Act has every right to inherit his property with his children. The Administration of Estates Law of Oyo State acknowledges this in similar terms. This ensures that a widow does not necessarily have to suffer the negative effects of customary law of succession, once it can be proved that she married under the Act. Thus, in the case of *Obusez v Obusez*, the deceased died intestate in 1988, leaving behind his wife, whom he married under the Marriage Act in 1972, and five children. The widow applied for letters of administration to administer the deceased's estate. It was granted to her and the children. The plaintiff, the twin brother of the deceased, challenged the decision, but lost at the Court of Appeal. On further appeal to the Supreme Court, it was held that 'it is not disputed that the deceased and his wife were married under the Marriage Act, but that prior to that marriage, both parties were subject to customary law.' Therefore, it follows that by virtue of the said marriage and upon the death of the deceased intestate, the provisions of the Administration of Estates Law of Lagos State (the deceased's domicile) become applicable. It is very clear from Section 49 (5) of the said law, that the intention of the law maker was that customary law be excluded in relation to the estate of persons in which the sub-section applies as explained above.

### 2.1.4. Other Discriminatory Practices

From the above, the Nigerian Constitution, which is the highest law of the country, states without any ambiguity, the right to freedom from discrimination (Odigie-Emmanuel 2010) in Section 42 and denounces discrimination on the grounds of gender, sex, circumstances of birth, etc.

Unfortunately, customary and religious laws continue to restrict women's rights, and the combination of a tripartite system of civil, customary, and religious laws often make

it extremely difficult to harmonise legislations and to remove discriminatory measures. Therefore, the status of the Nigerian woman is dependent on the legal system she is subject to. For example, the Islamic Sharia Law practiced in the Northern States of Nigeria reinforces customs that are unfavourable to women. For instance:

- The flogging of Bariya Magozu who was convicted for adultery in Zamfara State (Howard-Hassmann 2004).
- The death sentence by stoning passed on Safiya Hussaini Tunga Tudu by a Sharia Court in Sokoto for adultery while her male partner was allowed to go free (Human Rights Watch 2001).
- The death sentence passed on Amina Lawal in Funtua for having a child outside wedlock (Jare Ilelaboye 2003).

Apart from these religious laws, there are several regulations and government policies that are implemented contrary to Section 42 of the Nigerian Constitution, and Section 16 of the Convention on the Elimination of All forms of Discrimination against Women (CEDAW). For instance, in the recent case of *Dr. (Mrs) Priye Iyalla-Amadi v Nigerian Immigration Service (NIS)* (Human Rights Watch 2009) the Nigerian Immigration Service required married women to submit letters of consent from their husbands, as part of the requirements for processing passports for travel purposes. This requirement restricted the right to personal freedom and movement, as a married woman must obtain her husband's permission to travel. The Federal High Court of Nigeria did well in declaring such requirement unlawful.

*2.2. International Human Rights Conventions Binding on Nigeria Addressing Cultural/Discriminatory Practices against Women*

The Universal Declaration of Human Rights (UDHR) adopted in 1948, defines human rights broadly. While not much is said about women, Article 2 entitles all to "the rights and freedoms set forth in this Declaration, without distinction of any kind, such as race, color, sex, language, religion, political or other opinion, national or social origin, property, birth, or other status".

Discriminatory practices against women, which may culminate into violence against them, may involve rape, forced marriage, female genital mutilation, sex slavery, domestic violence, or denial of inheritance rights. Other situations, such as denying women decent education or jobs, leave them prey to abusive marriages, exploitative work, and prostitution. In addition to the general provisions in the UDHR, the rights of women not to be discriminated against, on the grounds recognized by International Human Rights Law, are stated in the Convention on the Elimination of all forms of Discrimination against Women (CEDAW), the African Charter on Human and Peoples' Rights (The African Charter), and the Protocol to the African Charter on the Rights of women in Africa (the Protocol) amongst other international conventions. National constitutions also include provisions of non-discrimination including Section 42 of the 1999 Nigerian constitution as stated above.

In summary, the Protocol of the African Charter on Human and Peoples' Rights on the Rights of Women in Africa takes into consideration the provisions of other international instruments on human rights that touch on women's rights, the need for equality, and freedom from discrimination. It also takes into consideration the particular circumstances of women in Africa, and their role in development. The protocol could certainly have been the impetus for change for Nigerian women, but unfortunately, a lot still needs to be done. The protocol was adopted on 11 July 2003 by the African Union in Maputo, Mozambique, to strengthen the promotion and protection of women's rights. The preamble highlights several considerations necessitating the protocol. These considerations include the recognition of Article 2 of the African Charter which enshrines the principle of non-discrimination. It includes Article 18, which calls on all states to eliminate discrimination against women.

By virtue of the protocol, Nigerian women are guaranteed the rights also included in the 1999 Nigerian Constitution. Some of the obligations of the Nigerian Government under

the protocol include ensuring that women enjoy the rights mentioned above through the following actions:

(a)  Enactment of appropriate legislation to combat all forms of discrimination, and specifically, to prohibit all forms of violence against women; to ensure prevention, punishment, and eradication of violence against women; to prohibit and punish all forms of genital mutilation; to guarantee that no marriage takes place by coercion. It should be between consenting adults. It also serves to ensure that men and women have the same rights during separation, divorce, and annulment of marriage; and to guarantee equal opportunity in work and career advancement.

(b)  Appropriate and effective education, administration, prohibition, protection, promotion, institutional, implementation and regulatory measures.

(c)  Integrating a gender perspective in policy decisions.

(d)  Modifying social and cultural patterns regarding the conduct of women and men through public education, information, and communication.

(e)  Positive action to promote participation of women in politics and decision-making.

(f)  Provision of effective remedies.

(g)  Ensuring full implementation at a national level.

(h)  Providing budgetary and other resources necessary for full and effective implementation.

Implementation of some of the above requirements may be difficult due to the requirement of Section 12 (1) of the 1999 Nigerian Constitution which provides: "No treaty between the Federation and any other country shall have the force of law except to the extent to which any such treaty has been enacted into law by the National Assembly". Fortunately, the Nigerian government has enacted the African Charter on Human and Peoples Rights as a domestic legislation and is cited as African Charter on Human and Peoples' Rights (Ratification and Enforcement) Act, Chapter A9, Laws of the Federation of Nigeria, 2004. Thus, the provisions of the African Charter can be adopted for the protection of women's rights and canvassed in the domestic courts if these rights are violated. Particularly Section 18 of the African charter states that: "The State shall ensure the elimination of every discrimination against women, and also ensure the protection of the rights of the woman and the child as stipulated in international declarations and conventions". The enforcement of the provisions of the African Charter has been emphasized by the Supreme Court of Nigeria in the case of *Abacha v Gani Fawehinmi* that:

> The individual rights contained in the Articles of the African Charter are justiciable in Nigerian courts. Thus, the articles of the Charter show that individuals are assured rights which they can seek to protect from being violated and if violated to seek appropriate remedies; and it is in the national courts such that protection and remedies can be sought and if the case is established and enforced.

This position has also been made clear in Nigeria in the case of *Ogugu v the state* where Bello CJN, in reference to the enforcement of the African Charter as to its human rights provisions within a domestic jurisdiction, observed as follows:

> Since the Charter has become part of our domestic laws, the enforcement of its provisions as all our other laws, fall within the judicial powers of the courts as provided by the constitution and all other laws relating thereto ... it is apparent... that the human and people's rights of the African Charter are enforceable by the several High courts, depending on the circumstances of each case and in accordance with the rules, practice, and procedure of each court.

The above statements by the Justices of the Supreme Court, the highest court in Nigeria, may turn the tide of discriminatory customs and policies as well as those that violate the rights of women and the girl-child.

*2.3. Provisions of the Nigerian Constitution 1999 and Legislative Enactments in Nigeria Relating to Cultural/Discriminatory Practices against Women*

Several attempts have been made by different non-governmental organizations and women's professional groups to bring bills before the National Assembly to enact CEDAW into domestic law, but these bills have never passed the required stages for enactment into law.

Furthermore, according to Article 1 of CEDAW, discrimination means:

Any distinction, exclusion, or restriction made on the basis of sex which has the effect or purpose of impairing or nullifying the recognition, enjoyment or exercise by women irrespective of their marital status, on a basis of equality of men and women of human rights and fundamental freedoms in the political, economic social, cultural, civil or any other fields.

Article 16 of CEDAW identifies the areas where discrimination against women can arise as follows:

State parties shall take all appropriate measures to eliminate discrimination against women in all matters relating to marriage and family relations and in particular shall ensure, on a basis of equality of men and women:

a.　　The same right to enter into marriage.
b.　　The same right to freely choose a spouse and to enter into marriage only with their free and full consent.
c.　　The same rights and responsibilities during marriage and its dissolution.
d.　　The same rights and responsibilities as parents, irrespective of their marital status, in matters relating to their children; in all cases the interests of the children shall be paramount...

Other pronouncements and arguments have been made in support of the enforcement of human rights treaties in Nigeria to ensure that these international treaties can be enacted as national laws. For instance, in the case of *Gani Fawehinmi v Abacha*, Gani Fawehinmi argued that:

Our customary laws should be weighed against this scale before they are applied by our courts. It should also be judicially noticed that it is against public policy not to recognize, implement, and enforce international treaties which Nigeria has signed and ratified.

Regrettably, there are challenges to the implementation of CEDAW and other provisions of statutes in Nigeria, due to the diversity of beliefs and cultural perspectives. The multi-layered and complex Nigerian indigenous systems continue to pose a challenge to the enforcement of women's rights as prescribed by the Nigerian Constitution and other international conventions.

It follows from the above that although the Constitution of Nigeria (1999) is recognized as binding on all Nigerians, its practical enforcement in changing some social issues seems to make it a toothless bulldog in the face of the overwhelming patriarchal structures with regard to women in Nigeria. It is important to note that laws in themselves are not sufficient to change the mindset of the people. Enforcement of such laws is important. However, the enactments of statutes such as the Administration of Estates Law cited above, the Child Rights Act, 2003, and other laws enacted by the Legislative arm of each state in the Federation of Nigeria, which are supported by Section 42 of the Nigerian Constitution, which court judgments tend to follow, adopt the Constitutional provision against discrimination and the common law principles of natural justice and equity, where a woman may suffer harm contrary to the fundamental human rights listed in chapter 4 of the 1999 Constitution. These rights include Sections 34 and 35, which provide for the right to dignity of the human person and right to liberty among other rights, particularly the right to life (Section 33). A court of law denouncing a cultural or administrative practice based on constitutional provisions and other domestic legislations, considering the diverse

nature of Nigeria's indigenous systems, can be said to be taking a step in ensuring that these discriminatory customs are gradually extinguished. Although CEDAW has not yet been passed into law in Nigeria, laws passed by the federal government and various states in Nigeria have to some extent incorporated these provisions, which constitute the foundation for defending women and children against discriminatory practices in Nigeria's indigenous systems.

The following laws are instructive, indicating progress in enforcing women's rights in Nigeria:

- The Violence Against Persons (Prohibition) Act (VAPP) was passed into law in May 2015. Section 6 of the Act criminalizes female genital mutilation, which is a cultural practice among many tribes in Nigeria.
- The Child Rights Act of 2003 was passed by the Federal Government of Nigeria and the different states, largely in the South and the East of Nigeria, have passed the Child Rights Law. This is a law through which the UN Convention on the Rights of the Child has been domesticated. The child right laws condemn child marriage and child labour which form part of the customary practices which can now be challenged as a result of this law and therefore seek to protect the right of the girl-child to education. Lagos state moved a step further to establish a Family Court on the 6th of June 2012 in accordance with the Child Rights Law of Lagos state to further secure the interests of children, especially their right to acquire an education, as opposed to the traditional belief that children can hawk items for sale in order to contribute to the economy of the household. This law is thus a positive development in challenging parents who, for economic gain, send their children out to sell wares, as opposed to sending them to school.
- The Federal Government also enacted an Act known as the Compulsory, Free Basic Education Act of 2004, to make primary education compulsory for all Nigerian children.
- Prohibition of Early Marriage Law in Kebbi and Niger states of 2002. This is a welcome development especially because these are Northern states where child marriage is widely practiced.
- Prohibition of Infringement of a Widow and Widower's Fundamental Rights Law, 2000. This is a state law enacted by the House of Assembly, Enugu State, (Eastern Nigeria) in 2001, to address the embarrassing notoriety the state had gained over the untold hardship and harmful traditional practices that widows and widowers are subjected to in the state.
- A law to prohibit girl-child marriage and female circumcision in Cross Rivers State 2000. This is a law which seeks to eradicate the practice of child marriage and female circumcision in Cross Rivers state in Nigeria.
- The National Policy on Women was also adopted in 2000. Under the policy, the Federal Government of Nigeria is expected to commit more resources to improving the lot of women in Nigeria by addressing the general and particular problems affecting them, and to ensure the mainstreaming of women's issues into all development programmes in a planned and sustainable manner. The policy includes a wide range of measures to eliminate discrimination in the areas of employment and labor, personal income tax, maternity leave, non-discriminatory terms and conditions of service, plus the provision of day care centers in offices and other general workplaces to be included in revised labor laws. Notably, the section on family, culture and socialization includes the following important strategies for implementation:

Reform customary marriage to eliminate huge dowry payment in cash and kind, and make small dowry payment a mere cultural symbolism; legislate against forced marriage and uphold law against child marriage in all Nigeria States; abolish by law in all Nigerian states all negative widowhood practices.

Others include: Review intestate succession law to ensure equal rights of inheritance of male and female children; implement the laws existing in some States against child mar-

riage; female circumcision (FGM), malpractices against widows and widowers, domestic violence and maltreatment, and ensure that other States enact similar laws; ensure that all Nigerian States adopt the Domestic Violence Prevention Bill (2005); create new legislations against all forms of domestic violence where none currently exists and grant women rights to family property (inheritance rights).

Since this policy was put in place, some laws have been enacted by some states such as the Child Rights Law of Oyo state and Lagos states. Nonetheless, enforcement is possible when the courts adjudicate on such matters to strike a final blow on discriminatory customs.

Additionally, a policy of this nature should cause a drastic change in the public and private sector by ensuring enforcement against perpetrators, particularly those who perpetrate acts of forcing young girls into marriage, female genital mutilation, and child labor while denying girls the right to an education amongst other discriminatory acts.

## 3. Comparative Analysis of the Laws of Other African Countries in Curbing Discrimination

In contrast, the Kenyan Constitution goes further than other African constitutions, by providing automatic application of international treaties to which Kenya has acceded (Ndulo 2011). Article 2(6) of the Kenyan Constitution provides that any treaty or convention ratified by Kenya shall form part of the law of Kenya under the Constitution. It further provides for the application of customary international law norms to Kenya. Clearly, international human rights norms prohibiting discrimination are applicable to Kenya. Similarly, the Constitution of the Republic of Malawi provides that 'any law that discriminates against women on the basis of gender or marital status shall be invalid...' It also obligates the government to take legislative measures that eliminate customs and practices that discriminate against women. In Section 10 (2), it further provides that 'in the application and development of customary law, the relevant organs of state shall have due regard to the principles and provisions of this Constitution.' Similarly, the Constitution of South Africa provides that 'the courts must apply customary law when that law is applicable, subject to the Constitution and any legislation that specifically deals with customary law.' Furthermore, the fundamental human rights provision of the Constitution of the Republic of Ghana guarantees the cultural rights and practices of the people, while still prohibiting 'all customary practices that dehumanize or are injurious to the physical or mental well-being of a person.' This modern approach is informed by the development of international human rights norms that outlaw discrimination. The Universal Declaration of Human Rights unequivocally prohibits discrimination.

In light of the above constitutions in other African countries, Nigeria may adopt similar ways of ensuring that customary law is subject to legislative enactments and the constitution. This may trigger substantial compliance so that customary law can be seen as subject to statutory law and the constitution, and may thus be interpreted in line with fundamental human rights of the citizens, as stated in the Nigerian Constitution. International human rights provisions can be included among the hierarchy of laws by ensuring the adoption into law of CEDAW, the Maputo Protocol and other relevant international conventions, in order to eradicate unwholesome and harmful cultural practices against women and girls.

## 4. Conclusions

This paper has discussed some discriminatory acts against women and how the Nigerian courts responded to these acts. The relevant international human right conventions were also highlighted, and unfortunately, though Nigeria is a signatory to these conventions, discriminatory customs and policies are still rife against the Nigerian woman. The cases cited emphasized that a girl-child is as important as a boy-child and should not be denied their rights to inherit what rightfully belongs to them after the demise of their father, and that practices such as child marriage and female genital mutilation should be eradicated as they are against human right laws and conventions. The discriminatory customs and practices discussed in this paper cover child marriage, inheritance and other

policies which tend to violate other rights such as a right to education. This paper thus proposes that such customs may be eradicated if the acts are reported and addressed by non-governmental organizations, human rights agencies, civil society organizations that may eventually direct these matters to the court of law. The state has an obligation to protect women's rights as the Nigerian constitution clearly states that no one should be discriminated against, either as a result of their sex or circumstances of their birth. The issue of child marriage is also a major custom or tradition that violates a girl's right to education and reproductive rights and eventually disempowers a woman when she is not properly educated as a result of early/child marriage. This runs contrary to CEDAW and other human right conventions and particularly against Section 21 of the Child Rights Act which prohibits child marriage and states that no person below the age of 18 years can enter into marriage.

The building of an egalitarian society cannot be achieved in a day. Women in Nigeria are discriminated against in all spheres of human endeavor. They are unprotected by the state, the society, and the family from discrimination despite the guarantee of equality by the constitution. It is therefore recommended that the National Gender Policy (2008–2013) should be fully implemented in accordance with its objectives which includes the implementation of national and international conventions. Secondly, the affirmative action policy derived from the National Policy on Women should be included in the Nigerian Constitution. Women in Nigeria constitute almost half of the national population. This numerical strength should be represented in Nigeria's public life, especially in elective positions to accelerate gender balance in all sectors, particularly towards the promotion of political rights, which will invariably affect the amendment of laws and decision-making. It is also imperative to upgrade the affirmative action policy in the National Policy on Women from an executive policy to a constitutional right to be included in the constitution.

Additionally, it is important that women and girls are educated about their rights so that they are aware that they can report any form of violence or discriminatory action against them. Although the act may have been committed by family members, a close relative, or even in the workplace, it is important that these acts are reported to the appropriate authorities. Publicizing these harmful discriminatory acts can protect women suffering under one harmful practice or the other, as well as others who may be silent victims. For instance, the sentence to death by stoning of Aminat Lawal was upheld by an Upper Sharia Court in Funtua. The case, taken to the Sharia Court of Appeal in the State capital by a non-governmental organization, was highly publicized through the media. This contributed to the prevention of the enforcement of the Sharia Law in the case, with human rights groups within and outside Nigeria rising up to her defense. Through the media, the international community became aware of the process and the potential for injustice (Ibrahim 2004).

Thus, efforts should be made to educate local groups and lawyers about widespread and deeply entrenched discrimination against women, and how this violates human rights law. The international diplomatic community in Nigeria should familiarize itself with local conditions and non-confrontational intervention, in the interest of judicial reform and adherence to equality under the law. As a result, the media coverage resulting in the publicity of the unfair Sharia decision may have had a positive impact on society, which resulted in the insistence on an adherence to human rights and the rule of law. This was one of the main arguments in the defense of Aminat Lawal.

**Author Contributions:** Writing original draft, F.O.A.; Formal analysis, S.O.E.; Review and editing, R.T. All authors have read and agreed to the published version of the manuscript.

**Funding:** This research received no external funding.

**Institutional Review Board Statement:** Not applicable.

**Informed Consent Statement:** Not applicable.

**Data Availability Statement:** Not applicable.

**Conflicts of Interest:** There is no conflict of interest among the authors.

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
