# Peer review of "Women’s Rights in Nigeria’s Indigenous Systems: An Analysis of Non-Discrimination and Equality under International Human Rights Law"

_socsci, doi:10.3390/socsci12070405_

Round 1

Reviewer 1 Report

Thanks for the opportunity to read this interesting paper. Below are my comments

My comment: The following statement in the paper needs a reference”

Although African states have ratified several human rights instruments protecting women's human rights, generally the severe political, economic and social difficulties facing African states have had a negative impact on the efforts to respect, protect and fulfill the human rights of women.

My comment: Please elaborate on the barriers

Some of the obstacles to the realisation of both the civil and political rights (CPR) and the economic, social and cultural rights (ESCR) of women are traditional practices and customs which disproportionately affect realization of thse rights as traditionally this category of rights has been often marginalised rather than prioritised

My comment: This (Mojekwu v Mojekwu) needs to be unpacked in more detail. I would suggest that the history of how each case came to courts be discussed in detail. Provide some background. 

For instance, in the case of Mojekwu v Mojekwu, Niki Tobi, Justice of the Court of Appeal, declared that the oli-ekpe customary law which states that a female child cannot inherit her father’s property in the absence of a male child is contrary to natural justice, equity and good conscience.

My comment: This needs some statistics to document the extent of the problem. At the moment, it is only a general statement.

This can result in exposure to sexually transmitted diseases including HIV as well as actual physical harm and the high health risks of early pregnancy by a yet-to-mature girl. This clearly violates the right to health and subsequently education, where the girl-child is forced to marry prematurely

My comment: Ethical and legal principles that such discrimination violates must be examined and analysed in detail. I think this should be done for each of the discussed cases here.

In the case of Motoh v Motoh the Court of Appeal held that the Native Law and Custom of Umuanaga Awka which discriminates against female children of the same parent and favours the male children who inherit the entire estate of their father to the exclusion of their female siblings, is repugnant to natural justice, equity and good conscience.

My comment: Which ethical and legal principles are to be defended by its adoption?

However, the presence of a statute or judgment of the court denouncing a cultural or administrative practice considering the diverse nature of Nigeria’s indigenous systems can be said to be a step in the right direction. Although CEDAW has not yet been passed into law in Nigeria, laws passed by the federal government and various states in Nigeria have incorporated these provisions to some extent which constitute the foundation for defending women and children against discriminatory practices in Nigeria’s indigenous systems

My comment: This is true but you need statistics or official data to back this up with references. Trends of some data over a recent time period would be best. I really think this is necessary, as you are right after claiming that there is drastic change in the enforcement.

The above stated Gender Policy reveals that there is an increase in awareness concerning women’s rights in Nigeria, but it can be said that the political will to prevail against the status quo is necessary for enforceability.

My comment: Gender equality indices must exist. I would suggest to mention a few in here and to provide an overview of the values over 1960 until today. This is mostly the time period when Africa countries became independent.

Eleanor Roosevelt and the Latin American women, who fought for the inclusion of sex in the declaration and for its passage, clearly intended that it would address the problem of women's subordination. According to Bunch and Polavarapu, female subordination/discrimination runs so deep that it is still viewed as inevitable or natural, rather than as a politically constructed reality maintained by patriarchal interests, ideology, and institutions. If violence and domination are understood as a politically constructed reality, it is possible to imagine deconstructing that system and building more just interactions between the sexes. Much of the abuse against women is part of a larger socio-economic web that entraps women, making them vulnerable to abuses which cannot be delineated as exclusively political or solely caused by states. Further, the assumption that states are not responsible for most violations of women's rights ignores the fact that such abuses, although committed perhaps by private citizens, are often condoned or even sanctioned by the states.

My comments: This section is more general than the Nigerian law section above. I think the UN and international law standards section should precede the section above

women not to be discriminated against, on the grounds recognised by International Human Rights Law, are stated in the Convention on the Elimination of all forms of Discrimination against Women (CEDAW), the African Charter on Human and Peoples Rights (The African Charter), and the Protocol to the African Charter on the Rights of women in Africa (the Protocol) amongst other international conventions. National constitutions also include provisions of non-discrimination including Section 42 of the 1999 Nigerian constitution as stated above.

My comment: I think one important point here will be the demonstration of the implementation of these commitments in the Nigerian legal and societal sphere. Can you demonstrate this with practical examples and challenges in implementation.

Implementation of some of the above requirements may be difficult due to the requirement of Section 12 (1) of the 1999 Nigerian Constitution which provides: “No treaty between the Federation and any other country shall have the force of law except to the extent to which any such treaty has been enacted into law by the National Assembly.” Thus, the above-stated provisions of the Protocol may not have the desired effect until the same has been enacted into law as a domestic legislation. However, some of these provisions have been enacted in different states and national legislations as outlined above

My comment: There is the need to provide a methodology to discuss how the author gathered the various reading materials for the study. What were the inclusion and the exclusion criteria used

Please clearly state all the specific objectives for the study

Author Response

I have attached the response to the reviewer one.

Reviewer 2 Report

Overall comments

I think there is great potential in this paper. The author(s?) demonstrate detailed knowledge of the subject and make compelling observations about the challenges of reconciling international and local legislative practices vis a vis women's rights. However, as an academic contribution, much more engagement with the state of the literature (especially recent works) on the subject is essential. The way that the paper is peppered with unnecessary detail of various laws and measures also distracts from the line of thinking presented by the author(s). However, I very much encourage a revision as I think there is enormous potential here!

Some specific points for each section below:

Abstract

The abstract is clear and interesting, with an appropriate level of detail.

Introduction

The introduction gives useful context to the reader as to the issues at hand. However, I suggest that the first and second paragraphs could benefit from further referencing and engagement with the literature, especially in relation to assertions made about: the translation of CEDAW at state level, sociocultural norms, the interaction between Nigerian courts and cultural belief systems.

Despite making reference at various points to the status of women’s rights, and their lower status, I think it’s important to include some actual data/statistics on gender inequality in Nigeria, eg girls’ educational access, women’s economic participation, health (including sexual and reproductive rights), political participation, interpersonal and familial power relations etc. This would help strengthen the rationale for the paper. I realise you mention the issue of child marriage later, but would be good to be able to situate and understand this in relation to the status of girls/women more generally.

It would be good to also know who is agitating for women’s empowerment in Nigeria  – and to what extent CEDAW and law reform is the means by which these claims are being made, or if activism is taking other forms within the country?

The majority of the final paragraph in the introduction feels beyond the scope of an introduction – I suggest it’s sufficient to state that the article concludes with observations and recommendations without detailing what these will be specifically.

Findings

There is a lot of ‘listing’ of items under CEDAW Articles 1 and 16, and various laws, measures, and policy areas. This information could be presented more concisely to make the same point, or even cut and replaced with a brief summary noting the most relevant points. At present it is distracting and disrupts the flow of text, meaning the reader loses the train of thought and argument of the author.

Bunch and Polavarapu (1990) references comprise most of the first paragraph of ‘International human rights conventions binding on Nigeria addressing cultural/ discriminatory practices against women’ – this section could be significantly shortened to just one or two cited points, especially given that the text is both 33 years old and not specifically about Nigeria.

The findings section also has too much text directly taken from the Protocol on the African Charter. Readers can access this themselves – please just summarise key points and link their relevance to the original observations you are presenting.

Conclusion 

The conclusion gives recommendations which are interesting, persuasive based on the evidence presented, and relevant. However, the conclusion needs to first summarise and bring together the key findings and implications from the different sections of the paper.

Author Response

Dear Editor,

I have attached the report to reviewer 2 for your attention.

Reviewer 3 Report

This paper is very interesting because it is able to narrate the issue of protecting the rights of women and girls in situations of legal diversity caused by the presence of various legal systems (state, customary, international) in society. The perspective of intersectionality also briefly appears in this paper. In order to improve the quality of this paper, several things need to be conveyed to the author.

First, on page 9 specifically the paragraph which mentions a number of laws and regulations, needs to be re-elaborated, as I cited here:

“A law to prohibit girl-child marriages and female circumcision in Cross Rivers State

2000.

The Child Rights Laws passed in 24 states in Nigeria which includes Abia, Akwa-

Ibom, Anambra, Bayelsa, Benue, Cross-River, Delta, Ebonyi, Edo, Ekiti, Enugu, Imo, Ka-

duna, Kogi, Kwara, Lagos, Niger, Ogun, Ondo, Osun, Oyo, Plateau, Rivers, and Taraba.

Street Trading Restriction Law 2004, Anambra State.

Schools (Parents, children and teachers) Law 2005, Rivers State.

Reproductive Health Services Law, 2003, Rivers State.

Trafficking in Persons (Prohibition) Law Enforcement and Administration Act, 2003.”

The elaboration should be made by the author because 1. the flow of narrative and argumentation can be developed and the reader understands the pluralistic context of legal products in Nigeria, 2. for readers who are not familiar with the context of the universe of law in Nigeria, they can understand briefly what articles or principles or values in these regulations are most relevant to the topic of the article.

Second, one of the references that is relevant and in my opinion good for use in analyzing both from the perspective of legal pluralism and feminist legal analysis, or gender and identity politics - on the construction of state law, custom and international conventions, is Anne Griffiths' article with the title of "Using Ethnography as a Tool in Legal Research: An Anthropological Perspective" in Banakar & Travers' Book : Theory and Method in Sociolegal Research, published by Onati Law School. Although your research is not meant to be legal research or research from a socio-legal perspective, Griffiths' writing is interesting to refer to in the context he wrote about how customary law, national law, and international conventions intertwine in the context of inheritance rights for women both in their position as children and widows. There is also the issue of gender-based discrimination discussed there.

Third,  Griffiths' article may also be useful to refer as one of the methodological foundations related to this draft, which seems to be fully using secondary material. I did not find any traces of empirical findings if the word empirical is interpreted narrowly and classically as having field research and the results are in the form of narrative direct interviews with informants. Of course, in a study using a socio-legal perspective, the presence of a judge's decision can be said to be something empirical as well because it is a product of the discourse that occurs in the courtroom. Thus, in my opinion, it is important for the author to add arguments and reading references related to how the empirical aspects in a social science research are broadly interpreted, in the context of the use of judge's decisions

Fourth, keywords need to be corrected, for example: gender construction, customary law, legal pluralism in Nigeria, etc.

Author Response

Dear Editor,

I have attached the report to reviewer 3.

Round 2

Reviewer 2 Report

Thank you for engaging with my suggestions and for clarifying the scope of the paper. I have no further suggestions and am pleased to recommend publication.